# A host–guest approach to combining enzymatic and artificial catalysis for catalyzing biomimetic monooxygenation

Liang Zhao [1], Junkai Cai[1], Yanan Li[1], Jianwei Wei[1] & Chunying Duan [1,2] ✉

Direct transfer of protons and electrons between two tandem reactions is still a great challenge, because overall reaction kinetics is seriously affected by diffusion rate of the proton and electron carriers. We herein report a host–guest supramolecular strategy based on the incorporation of NADH mimics onto the surface of a metal-organic capsule to encapsulate flavin analogues for catalytic biomimetic monooxygenations in conjunction with enzymes. Coupling an artificial catalysis and a natural enzymatic catalysis in the pocket of an enzyme, this host–guest catalyst–enzyme system allows direct proton and electron transport between two catalytic processes via NADH mimics for the monooxygenation of both cyclobutanones and thioethers. This host–guest approach, which involves the direct coupling of abiotic and biotic catalysts via a NADH-containing host, is quite promising compared to normal catalyst–enzyme systems, as it offers the key advantages of supramolecular catalysis in integrated chemical and biological synthetic sequences.

[1] State Key Laboratory of Fine Chemicals, Dalian University of Technology, 116024 Dalian, People's Republic of China. [2] Zhang Dayu School of Chemistry, Dalian University of Technology, 116024 Dalian, People's Republic of China. ✉email: cyduan@dlut.edu.cn

Catalyst–enzyme coupled systems that combine artificial and natural enzymes in one pot to reproduce the natural prototypes of enzymatic processes[1,2] have emerged as attractive and competitive ways to match the high efficiency and selectivity of enzymatic systems[3–6]. Significant advances have been made in the integration of individual redox enzymes into proof-of-principle photocatalytic systems to drive a litany of complex organic redox transformations efficiently with sustainable feedstocks[6]. However, the combination of abiotic and biotic catalysts is often challenging since it goes beyond the ability of synthetic complexes to repeat catalytic behavior of natural enzymes[7–10], and requires the catalysts in semibiologically systems to be mutually compatible when operating under similar conditions and at similar rates[4,11]. Notably, such compatibility between abiotic and biotic components also requires the fine matching of the electron transfer pathways in each of the tandem transformations and often requires stoichiometric equivalents of expensive reducing cofactors, such as NADH (reduced nicotinamide adenine dinucleotide), to mediate the enzymatic reaction[12–14]. We think that the installation of active sites of cofactors i.e. NADH into the backbone of metal-organic capsules[15,16] would be a distinguished strategy for the development of artificial catalytic platforms to work with natural enzymes. We envision that the NADH mimics located on the surface of a capsule would allow the NADH mimics to receive protons and electrons directly from enzymatic processes when the capsule enters the catalytic domain of the enzyme[17,18], eliminating the diffusion processes of cofactors in catalysis. The unique structural restrictions and proximity effects of the substrates in the supramolecular host would also increase effective collisions for the effective biomimetic transformations involving the NADH mimics[19–24], and the spatial isolation of artificial catalysis and enzymatic catalysis has been demonstrated to prevent mutual interference[4,17,25,26].

On the other hand, flavin-dependent enzymes, which are ubiquitous in nature, provide sufficient activation to insert oxygen into substrates with definite reaction mechanism that involves the reduction of flavin adenine dinucleotide (FAD) by NADH to yield a reduced $FADH_2$ and an interaction between a rapidly formed peroxo species, FAD-OOH and the substrate[27–29]; thus, we hypothesize that the supramolecular catalytic system with the NADH-modified capsule should be applicable for simulating flavin-dependent enzymes with different enzymes attributed to its inherent host–guest behavior. Considering that flavin-dependent enzymes always hold the flavin catalytic active site and substrate a close proximity (ca. 10 Å) and stabilize the reaction intermediates through supramolecular forces[30,31], the supramolecular systems that enforced cofactor, catalyst and substrate together would be beneficial to the effective biomimetic transformations[32,33]. Moreover, this promising approach for coupling the artificial enzyme and the natural enzyme into one pocket of the enzyme would allow direct proton and electron transport between the two enzymatic processes through the proton and electron transport channels via NADH mimics (Fig. 1). The fine synergy of the reactions that occur inside and outside of the artificial host and the pocket of the natural enzymes would make the NADH-containing host–guest complex an attractive target for the development of supramolecular hosts for catalysis.

Herein, by installing NADH mimics into the backbone of the bistridentate ligand, we develop a metal-organic capsule that can encapsulate a simple flavin model, riboflavin tetraacetate (RFT)[34], for the monooxygenation of cyclobutanones and thioethers in tandem with enzymes. The ingenious design of host–guest supramolecular system well-controls the multiple electron transfer process during catalysis, allowing the tandem reaction to occur smoothly with high efficiency and selectivity even under a low intensity light (3 W household fluorescent lamp), which offers the key advantages of supramolecular catalysis in integrated chemical and biological synthetic sequences.

## Results

**Preparation and structural characterization of the hosts**. A ligand ($H_2$ZPA) containing a dihydropyridine amido (DHPA) mimic similar to the active site of an NADH model with a pendant C(O)–NH group as potential hydrogen-bonding sites was synthesized by the Schiff base reaction of 2-formylpyridine and 4-(3,5-di(hydrazine-carbonyl)-4-phenyl-pyridin-1(4H)-yl)-N-methyl-benzamide in ethanol (Supplementary Fig. 1). Ligand $H_2$PMA was synthesized by the Suzuki cross-coupling reaction of dimethyl 5-iodoisophthalate and (3,5-dimethoxycarbonylphenyl) boronic acid pinacol ester, and the procedure described for ligand $H_2$ZPA was used here, except a benzene ring substituent was present in the active sites of the NADH model (Supplementary Fig. 2). Stirring equimolar solutions of ligand $H_2$ZPA with the NADH model or $H_2$PMA and $Zn(BF_4)_2 \cdot H_2O$ gave macrocycles Zn–ZPA and Zn–PMA in yields of 60% and 52%, respectively, at room temperature.

The coordination of the ligands to metal ions was indicated by the disappearance of the imine proton C(O)–NH signal at 11.60 ppm and the significant downfield shifts (0.12–0.17 ppm) of the tertiary carbon protons (N=CH) and pyridine ring protons in the spectrum of the complexes with respect to the spectrum of the free ligand (Supplementary Figs. 9 and 13). Specifically, Q-TOF-MS analysis exhibited the expected isotopic patterns of $[H_3Zn_4(ZPA)_4]^{3+}$ species at $m/z = 864.5278$ for Zn–ZPA (Supplementary Fig. 7) and of $H_nZn_4[(PMA)_4]^{n+}$ ($n = 2, 3$) species at $m/z = 759.1381$ and $1138.1724$ for Zn–PMA (Supplementary Fig. 8) in $CH_3CN$, respectively, showing the integrity of both capsules in $CH_3CN$. And there were the same results in the mixed solvent system of $CH_3CN$/PBS (2:1), indicating that the molecular capsules are suitably stable in solution.

Single-crystal structural analysis revealed that Zn–ZPA and Zn–PMA were both $M_4L_4$ tetranuclear macrocyclic capsules. Zn–ZPA crystallizes in the tetragonal space group $P\bar{4}n2$, showing one-fourth of a molecule macrocycle in the asymmetric unit, a crystallographic $C_2$-axis passing through the center of tetranuclear macrocycle, and the alternating connection of four ligands containing the central DHPA fragments and four metal zinc ions via $S_4$ symmetry (Fig. 2a–c and Supplementary Figs. 3 and 4)[35]. Each zinc ion shows an octahedral coordination geometry with the Zn center chelated by two ONN tridentate chelators from different ligands positioned in a mer configuration. Each ligand linearly bridged two adjacent zinc ions to maintain an average Zn…Zn separation of 8.9 Å, suggesting that the cavity of Zn–ZPA is sufficiently large to encapsulate RFT (ca. 7.7 Å) to form the binding domains and ensure a very close proximity (<10 Å) between the substrate and the flavin peroxide before the reaction[28]. The two DHPA moieties on the opposite side are positioned parallelly on either side of the capsule windows and all the active H-atoms on the DHPA moieties point toward the interior of the capsule, which favors the interaction between the NADH models and RFT within Zn–ZPA to produce $RFTH_2$. The benzamide groups act as binding sites, and they are in the same arrangement as the DHPA moieties, further ensuring that Zn–ZPA and RFT could form a stable host–guest system[36]. In addition, the enriched hydrogen-bonding sites located on the capsule would stabilize the flavin semiquinone radicals and flavin peroxides formed during the catalytic reaction through intramolecular hydrogen bonding[37]. The distribution of the NADH mimics on the surface of Zn–ZPA connects the internal and external environment of the capsule, ensuring the rapid

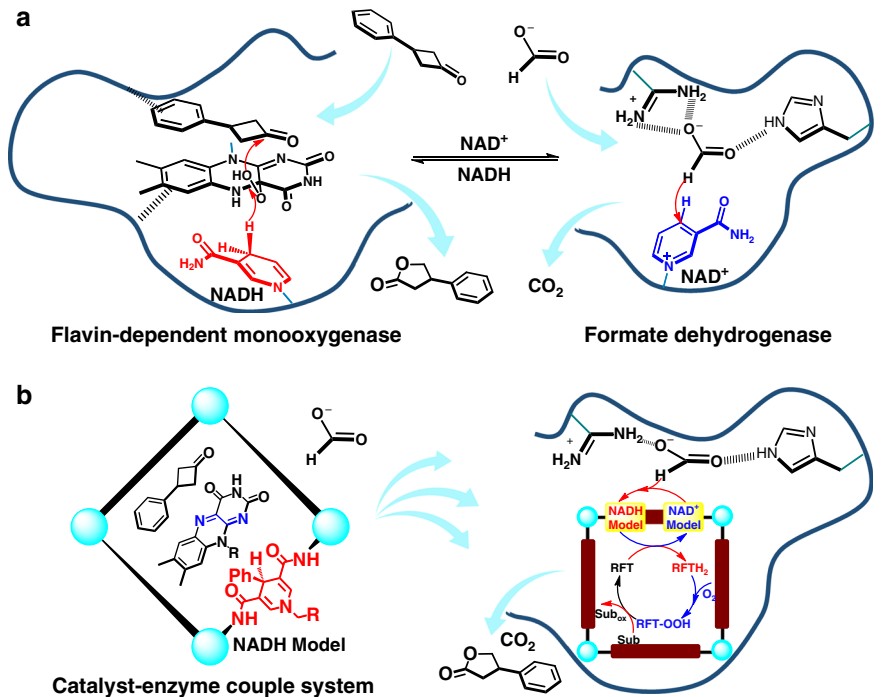

**Fig. 1 Schematic of natural enzymatic system and catalyst–enzyme couple catalytic system. a** Multienzyme cascade comprising flavin-dependent monooxygenase and formate dehydrogenase connected by diffusion of the nicotinamide cofactor (NADH/NAD+). **b** Superstructures comprising NADH-modified metal-organic capsule and enzyme achieve the coupling of artificial and enzymatic catalysis in situ without diffusion.

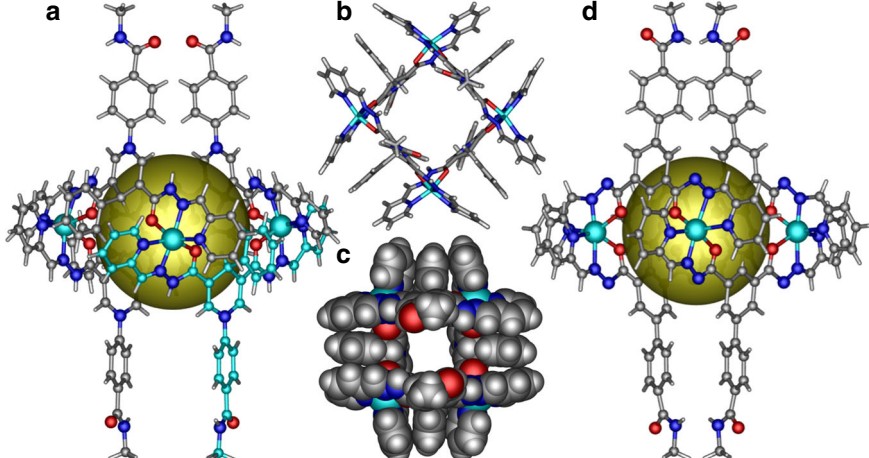

**Fig. 2 Crystal structures of the molecular macrocycle hosts. a** Crystal structure of the macrocycle Zn–**ZPA** showing the coordination geometry of the zinc ions, the pocket and the distribution of the NADH-mimicking active sites and amide groups on the H₂**ZPA** ligands. **b** Top view and **c** space-filling pattern of the macrocycle Zn–**ZPA** showing the resulting confined space and the open channels. **d** Crystal structure of the macrocycle Zn–**PMA**. Zn: cyan, O: red, N: blue, C: gray and H: white. Solvent molecules and counter ions are omitted for clarity.

transmission of protons and electrons to turn over the catalytic cycle.

The Zn–**PMA** macrocycle crystallizes in the $P\bar{4}$ space group and exhibits ideal $S_4$ symmetry, and its zinc ions and ligands are located on the crystallographic $C_2$ axis (Fig. 2d and Supplementary Figs. 5 and 6). Zn–**PMA** possesses the same structural features as Zn–**ZPA** except the benzene ring substituent in the active site of the NADH model and an average Zn···Zn separation of 8.3 Å. Because Zn–**PMA** exhibits almost the same coordination structure as Zn–**ZPA**, Zn–**PMA** was considered an ideal reference compound for Zn–**ZPA** to investigate the encapsulation of flavin analogs into the NADH-modified metal-

organic capsules for the collaborative catalysis of monooxygenation reactions with enzymes.

**Monooxygenation and host–guest interactions.** As one of the most studied Baeyer–Villiger substrates for the conversion of cyclobutanone derivatives to furanone derivatives, 3-phenyl-cyclobutanone (**1a**) was chosen as an ideal model substrate for our study. Upon addition of Zn–**ZPA** to a solution of RFT (10.0 μM), a significant quenching of the emission at 535 nm was observed, which could be attributable to photoinduced proton-coupled electron transfer (PCET) from the NADH module to the excited

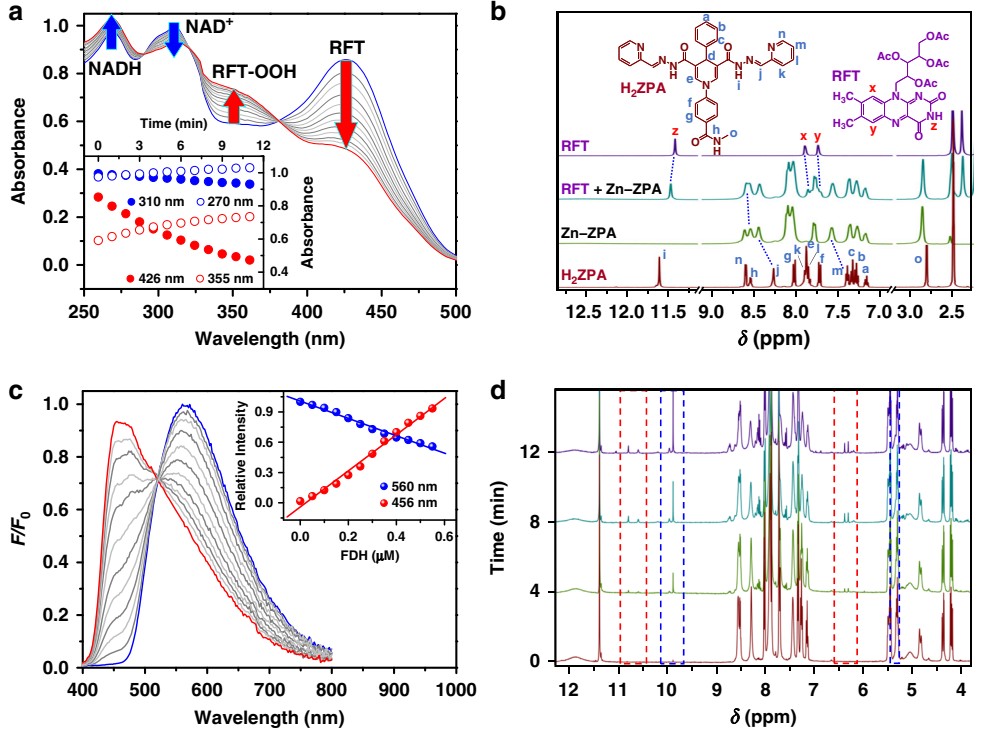

**Fig. 3 Characterization of the interactions between Zn–ZPA and RFT or FDH. a** UV-Vis spectrum of Zn–ZPA (10.0 μM) and RFT (10.0 μM) in a CH₃CN solution in the presence of oxygen under the irradiation at 455 nm. The inset shows the changes in the absorption at 270, 310, 355 and 426 nm as a function of irradiation time. **b** Partial ¹H NMR spectra of the free ligand H₂ZPA (red), Zn–ZPA (green), an equimolar mixture of Zn–ZPA and RFT (cyan) and RFT (purple) in DMSO-d₆. **c** Family of luminescence spectra of Zn–ZPA (10.0 μM) in CH₃CN/H₂O (2:1) upon the addition of FDH (total 0.5 μM). The inset shows the changes at 460 and 565 nm. **d** Partial ¹H NMR spectra recorded during the reaction of an equimolar Zn–ZPA and RFT in the absence of oxygen under the irradiation at 455 nm in DMSO-d₆.

state RFT* (Supplementary Fig. 17). However, the luminescence lifetime of the RFT solution (10.0 μM) containing Zn–ZPA (10.0 μM) at 535 nm decayed similarly to that of free RFT (Supplementary Fig. 22). These results suggested that PCET occurred within the pocket of Zn–ZPA via a pseudointramolecular pathway[38]. Irradiation of the aforementioned solution led to a gradual decrease in the absorption at 426 nm and enhanced absorption at 355 nm in the UV-Vis spectra (Fig. 3a), which can reasonably be attributed to a reduction of the RFT. In addition, the diminishing intensity at 310 nm with the increasing intensity at 270 nm suggested that the NADH models on Zn–ZPA were converted to NAD⁺ groups, indicating the occurrence of hydride transfer processes resembling FAD-dependent oxidases in biological systems[39,40].

In this case, irradiation of a CH₃CN/EtOAc/H₂O (8:1:1) solution containing Zn–ZPA (2.0 mM), RFT (4.0 mM) and **1a** (2.0 mM) with a typical 3 W household fluorescent lamp in the presence of oxygen for 12 h resulted in an 85% yield (Table 1, entry 1, racemates, no further separation for the two enantiomers)[41]. Notably, the reduced nicotinamide adenine nucleotide (NADH, 8.0 mM), the Hantzsch esters (HEHs, 8.0 mM)[42] or the ligand (H₂ZPA, 8.0 mM), as alternatives to Zn–ZPA, could not efficiently insert an oxygen atom into the substrate **1a** (Table 1, entries 2–4). Obviously, the confined space effect of compound Zn–ZPA plays the crucial role in completing monooxygenation. Control experiments demonstrated that Zn–ZPA containing NADH mimics, RFT, oxygen and light were all required for monooxygenation (Table 1, entries 5–8). Control experiments performed with Zn–PMA using the same conditions that were used with Zn–ZPA demonstrated that the absence of active sites

in Zn–PMA prevented monooxygenation (Table 1, entry 9). These results combined with the results of the H₂ZPA catalyzed reaction indicated that the encapsulation of the RFT and substrate catalyzed hydride transfer processes, leading the reaction toward the targets.

The formation of host–guest complexation species and the distribution of NADH mimics on the surface of the capsules enable the direct communication between the reactions within the capsule and on the outside of the capsule, which is the basis for the biocatalysis within the pocket of the capsules and the regeneration of the NADH mimics outside the pocket of the capsules. The theoretical 'docking study' results of Zn–ZPA ⊃ RFT showed the existence of multiple hydrogen-bonding between the hydrazides on the capsule Zn–ZPA and the carbonyl group and acetyls of RFT (Fig. 4a and Supplementary Fig. 36), suggesting that many hydrogen-bonding sites on Zn–ZPA played an auxiliary role in trapping the guest molecule RFT. Upon addition of Zn–ZPA (10.0 mM) to RFT (10.0 mM), the ¹H NMR signal of the active Hd atom in the DHPA moieties showed a slight downfield shift (Supplementary Fig. 10), suggesting the occurrence of interactions between the RFT by NADH mimics. In addition, the signals associated with active protons Hz on RFT and Hh on Zn–ZPA shifted downfield, further confirming that the formation of multiple hydrogen bonds between RFT and Zn–ZPA shortened the distance between active sites effectively (Fig. 3b)[43]. ¹H NMR monitoring of the hydride transfer processes between the RFT and Zn–ZPA showed a gradual increase in the area of the peaks of RFT at ~6.33, 6.40, 10.62 and 10.82 ppm belonging to RFTH₂ and at 9.91 belonging to the NAD⁺ models on

**Table 1 The Baeyer–Villiger oxidation of 3-phenylcyclobutanone catalyzed by RFT with different cofactors in the presence of oxygen.**

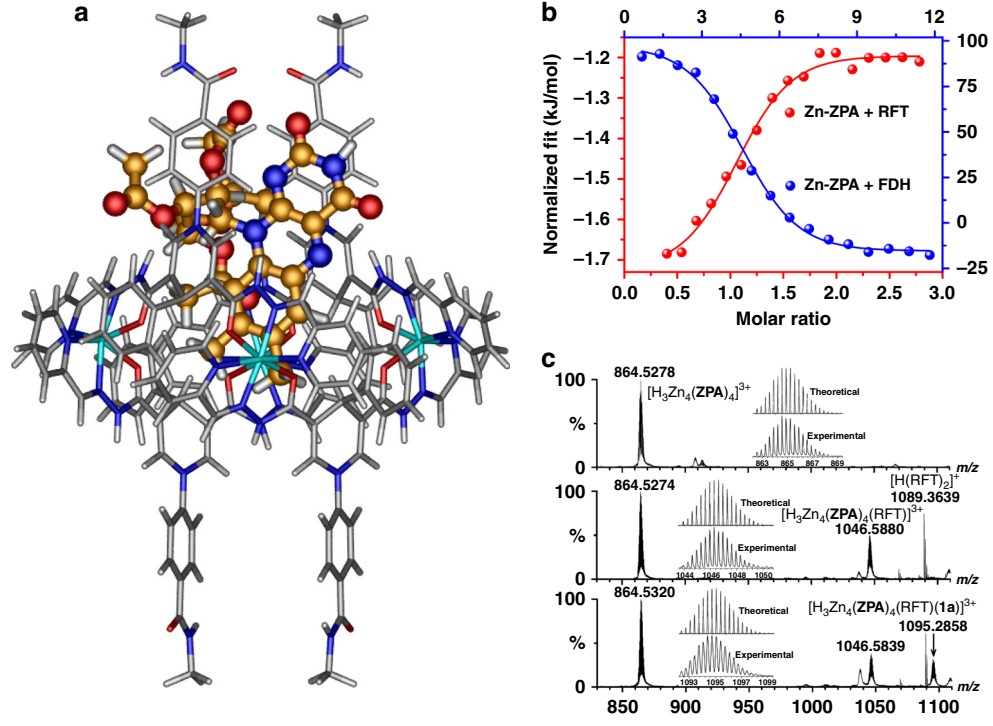

| Entry | Cofactor | Loading (mM) | RFT (mM) | Oxidant | Yield (%) |
|---|---|---|---|---|---|
| 1 | Zn-**ZPA** | 2.0 | 4.0 | $O_2$ | 85 |
| 2 | NADH | 8.0 | 4.0 | $O_2$ | 7 |
| 3 | HEH | 8.0 | 4.0 | $O_2$ | 5 |
| 4 | H$_2$**ZPA** | 8.0 | 4.0 | $O_2$ | 3 |
| 5 | — | — | 4.0 | $O_2$ | Trace |
| 6 | Zn-**ZPA** | 2.0 | — | $O_2$ | Trace |
| 7 | Zn-**ZPA** | 2.0 | 4.0 | — | Trace |
| 8[a] | Zn-**ZPA** | 2.0 | 4.0 | $O_2$ | Trace |
| 9 | Zn-**PMA** | 2.0 | 4.0 | $O_2$ | Trace |

Reaction conditions: CH$_3$CN/EtOAc/H$_2$O (8:1:1), **1a** (2.0 mM), fluorescent lamp (3 W), 37 °C, 12 h. Yields were determined by integration of the $^1$H NMR peaks and comparison to internal standards.
[a]Without light.

**Fig. 4 Characterization of the host–guest behavior between Zn–ZPA and RFT or FDH. a** Representation of the encapsulation of Zn-**ZPA** and RFT computed by the theoretical 'docking study' calculations. **b** ITC experiments on Zn-**ZPA** upon the addition of RFT or FDH showing the formation of host–guest complexes in CH$_3$CN/H$_2$O (2:1). **c** ESI-MS spectra of Zn-**ZPA** (top), Zn-**ZPA** (middle) following the addition of 1 equiv of RFT and Zn-**ZPA** (bottom) following the addition of 1 equiv of **1a** and RFT in CH$_3$CN solution. The insets show the measured and simulated isotopic patterns at $m/z = 864.5278$, 1046.5880 and 1095.2858.

Zn–**ZPA**, whereas the peaks of the NADH active sites on Zn–**ZPA** at ~5.35 ppm gradually disappeared as the reaction progressed in the absence of oxygen (Fig. 3d)[44]. After exposing the aforementioned solution to oxygen, the peaks attributable to RFTH$_2$ gradually disappeared, and at the same time, a new peak gradually formed at ~10.2 ppm (Supplementary Figs. 11 and 12), which could be interpreted as a signal of flavin hydroperoxide that could also detected by potassium starch iodide test

(Supplementary Fig. 29)[45]. The spectral results suggested that the short distance between the catalytic centers and cofactors due to the formation of host–guest species was the required for hydride transfer to form RFTH$_2$, which could then react rapidly with oxygen to catalyze monooxygenation.

Isothermal titration calorimetry (ITC) experiments showed that upon the addition of RFT, Zn–**ZPA** gave enthalpic and entropic changes of $\Delta H = -0.58$ kJ mol$^{-1}$ and $-T\Delta S = 26.81$ kJ mol$^{-1}$,

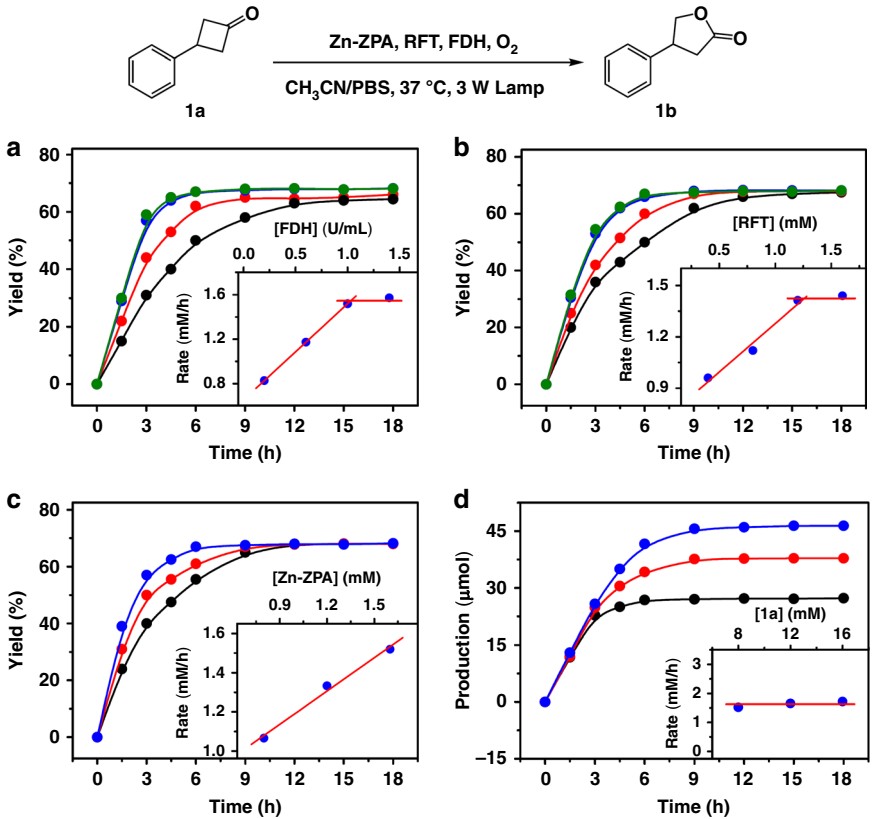

**Fig. 5 Kinetics of the Baeyer–Villiger oxidation. a** Yield of **1b** as a function of the concentration of FDH with RFT (1.6 mM), Zn-**ZPA** (1.6 mM) and **1a** (8.0 mM) remaining fixed. **b** Yield of **1b** as a function of the concentration of RFT with FDH (1 U mL$^{-1}$), Zn-**ZPA** (1.6 mM) and **1a** (8.0 mM) remaining fixed. **c** Yield of **1b** as a function of the concentration of Zn-**ZPA** with FDH (1 U mL$^{-1}$), RFT (1.6 mM) and **1a** (8.0 mM) remaining fixed. **d** Yield of **1b** as a function of the concentration of **1a** with FDH (1 U mL$^{-1}$), Zn-**ZPA** (1.6 mM) and RFT (1.6 mM) remaining fixed; the reaction contained HCOONa (16.0 mM) and was conducted in a CH$_3$CN/PBS (2:1) solution under a 3 W fluorescent lamp in the presence of oxygen.

respectively, and a Gibbs-free energy change of $\Delta G = -27.39$ kJ mol$^{-1}$ with an association constant of $K_a = 6.23 \times 10^4$ M$^{-1}$ for the Zn-**ZPA** ⊃ RFT complex (Fig. 4b and Supplementary Fig. 24)[46]. The formation of the host–guest species was largely driven by entropic considerations, and this allowed the reproduction of the binding domain in the flavin-dependent enzyme[28] due to the activation of binding, which the markedly enhanced reaction rates by four orders of magnitude in such a monooxygenation[47].

The supramolecular system also increased the number of effective collisions for the catalytic processes within the interior of the capsules due to the confinement effects, separating the redox events and avoiding mutual interference. When RFT (0.1 mM) was added into the solution of Zn–**ZPA** (0.1 mM), a new peak appeared at $m/z = 1046.5880$, and it showed natural isotopic abundances consistent with the formation of the 1:1 stoichiometric host–guest species [H$_3$Zn$_4$(**ZPA**)$_4$(RFT)]$^{3+}$ (Fig. 4c). Fortunately, the simultaneous addition of the reactant (3-phenylcyclobutanone, **1a**, 0.1 mM) and RFT (0.1 mM) into the solution of Zn–**ZPA** (0.1 mM) caused the appearance of a new peak at $m/z = 1095.2858$, which was assigned to [H$_3$Zn$_4$(**ZPA**)$_4$(RFT)(**1a**)]$^{3+}$ (Fig. 4c), and this peak confirmed that RFT and substrate **1a** were coencapsulated into the cavity of Zn–**ZPA** in a 1:1:1 host–catalyst–substrate fashion. In addition, host–guest species [H$_2$Zn$_4$(**PMA**)$_4$(RFT)]$^{2+}$ could also be formed, as indicated by the peak at $m/z = 1411.2938$. Clearly, hydride transfer between the flavin analog and the NADH mimics in the supramolecular system dominates the photocatalytic transformation, and the NADH mimics on the surface of the capsule allow

interaction with the enzymatic transformation within the pocket of natural enzymes to achieve the tandem conversion.

**Supramolecular catalysis in collaboration with enzymes.** The NADH mimics, which are located on the surface of the Zn-**ZPA** macrocycle, provide transport channels for protons and electrons to combine the monooxygenation and in situ regeneration of the NADH models. A CH$_3$CN/PBS (2:1) solution containing substrate **1a** (8.0 mM), Zn–**ZPA** (1.6 mM), RFT (1.6 mM), formate dehydrogenase (FDH, 1 U mL$^{-1}$) and HCOONa (16.0 mM) directly afforded product **1b** in 68% yield upon irradiation with a 3 W fluorescent lamp, and supplementation with FDH (total 2 U mL$^{-1}$) increased the yield to 87% in next 9 h (Supplementary Fig. 34). Moreover, when increasing **1a** concentration from 8.0 to 16.0 mM, the formation of **1b** showed a linear growing trend, whereas only a yield of 15% was observed in the absence of FDH (Supplementary Table 5), indicating that the enzyme was necessary to complete the overall catalytic cycle reaction system. And the peak attributed to Zn-**ZPA** could still be detected by ESI-MS after the reaction. It is postulated that the integrity of Zn-**ZPA** did not change during the monooxygenation processes.

When the concentration of FDH was changed (Fig. 5a), the total yield of **1b** did not vary, but the initial rate of the catalytic process increased linearly with increasing FDH concentration (from 0.2 to 1.0 U mL$^{-1}$), but remained essentially unchanged at concentrations above 1.0 U mL$^{-1}$. The results indicated that the enzymatic reaction regulated the rate of the NADH model regeneration on Zn–**ZPA**, and this reaction matched the reduction of flavin without harassing the catalytic processes

within the cavity of Zn–**ZPA**. Moreover, the overall catalytic monooxygenation exhibited pseudo-zero-order kinetic behavior, generally consistent with a Michaelis–Menten mechanism, and we envisioned that there was an interaction between FDH and the capsules relevant to Zn–**ZPA** combined with the catalytic domain of FDH. The luminescence titration of Zn–**ZPA** (10.0 μM) showed the quenching of the luminescence of Zn–**ZPA** at 565 nm along with the emergence of a new blueshifted peak at 460 nm upon the addition of FDH (0.5 μM) (Fig. 3c). The observed quenching and enhancement effects seen in the parent peaks were attributed the recognition between Zn–**ZPA** and FDH[48], the linear decrease of luminescence intensity for Zn–**ZPA** and the linear increase of peak at 460 nm also indicated an efficient binding process between the FDH and Zn–**ZPA**. With the addition of FDH (total 1.0 μM) to Zn–**ZPA** (10.0 μM), the intensity of the UV-Vis absorption at 300 nm significantly increased and that at 436 nm clearly decreased with isosbestic points at 417 nm and 469 nm (Supplementary Fig. 23). Together, the experimental UV-Vis and fluorescence spectra suggested the generation of host–guest species and further verified the noncovalent link between Zn–**ZPA** and the enzyme as well as the resulting weaker planar conjugation on the capsule due to the contraction of the enzyme structure[49,50]. The straightforward calculations of the thermodynamics of Zn–**ZPA** binding to FDH were conducted based on ITC (Fig. 4b and Supplementary Fig. 25). Good agreement was obtained in the fitting of the experimental isotherms, providing the association constant $K_a = 3.98 \times 10^6$ M$^{-1}$ and a change in the Gibbs-free energy of $\Delta G = -37.67$ kJ mol$^{-1}$, reflecting the high affinity and stability of the Zn–**ZPA**/FDH complex[51,52]. Therefore, Zn–**ZPA** could be attached to the catalytic domain of FDH, allowing them to directly participate in the enzymatic reaction, and unlike in natural enzyme systems, the consumed NADH mimics could be recovered in situ without diffusion, forming a closed loop of electrons and protons.

Catalytic reactions were also performed by varying the concentration of RFT while maintaining the other parameters (Fig. 5b). The magnitude of the effect of flavin on the overall catalytic processes decreased as the host–guest species gradually became saturated, especially when the RFT concentration ranged from 1.2 mM to 1.6 mM. The initial turnover frequency increased linearly when the concentration of Zn–**ZPA** was varied from 0.8 to 1.6 mM (Fig. 5c). Obviously, the formation of the host–guest complexes was essential for catalysis. The initial rate of the monooxygenation reaction remained constant when only the concentration of substrate **1a** was changed (Fig. 5d); clearly, the high concentration of **1a** maintained the zero-order catalysis for a longer time and the rate of the reaction was dependent on the concentration of the host–guest species rather than on the concentration of the substrate[53,54].

Interestingly, the addition of the nonreactive species adenosine triphosphate[55] (ATP) (8.0 mM) to a solution containing RFT (1.6 mM)/Zn–**ZPA** (1.6 mM)/FDH (2 U mL$^{-1}$) gave a yield of only 18% under the same reaction conditions (Supplementary Table. 5). Importantly, the addition of ATP (50.0 μM) to a solution containing RFT (10.0 μM) and Zn–**ZPA** (10.0 μM) resulted in a remarkable recovery of the emission intensity of RFT without a shift in the peak position (Supplementary Fig. 20). The addition of ATP (50.0 μM) to a solution of RFT (10.0 μM) did not affect the emission of RFT. The recovered emission of RFT revealed the extrusion of the RFT that was encapsulated in the pocket of Zn–**ZPA** by ATP. In this case, the competitive inhibition behavior suggested that the reaction occurred within the pocket of Zn–**ZPA** and that the host–guest system was superior for the catalytic monooxygenation (Supplementary Figs. 14, 18 and 19). The NADH-modified capsule Zn–**ZPA**

locked in the pocket of the natural enzyme eliminated diffusion and migration of the cofactors between the two redox loops. Therefore, the monooxygenation inside Zn–**ZPA** worked synergistically with enzymatic catalysis and the HCOO$^-$ oxidation reaction outside Zn–**ZPA** are simultaneous with mutual conversion between NADH models and NAD$^+$ model on the capsule surface. Such an efficient catalytic result for the tandem process indicated that the rationally designed supramolecular catalytic systems were superior, as they eliminated the inherent communication barrier and interference between artificial catalysts and enzymes.

From a mechanistic viewpoint, this catalyst–enzyme coupled system based on the NADH-modified metal-organic capsule not only maintained the traditional advantages of confinement in supramolecular systems but also offered enhanced proton and electron transfer inside and outside the capsule through cofactor channels. The well-designed molecular squares, which have amide-enriched hydrogen-bonding sites, stacking interaction sites and hydrophobic pockets, could simultaneously encapsulate RFT and the substrate (Fig. 6). Inside the capsules, the well-matched shape of the capsule forces the active H-atoms of DHPA, which are positioned toward the pocket of the square, to be in a close proximity to the RFT and to both guests. This host–guest complexation mode ensures that the transfer hydrogenation between RFT and the NADH mimics and the monooxygenation of cyclobutanones occur efficiently to produce cyclobutyrolactones and NAD$^+$ mimics under weak irradiation, reminiscent of enzymes. In the presence of the enzyme, the capsule carries the entire monooxygenation system into the catalytic domain of the FDH enzyme, eliminating the effects of cofactor diffusion on the reaction order during catalysis and simplifying the multisubstrate process into a single substrate process. The theoretical 'docking study' suggested that the pendant amide bonds on the capsule are vertically connected to the enzyme protein so that form a transparent channel with the original opening window of the enzyme in line with expectations, ensuring the normal exchange of substances inside and outside the host–guest supramolecular systems (Supplementary Fig. 37). The NAD$^+$ models generated in the monooxygenation can be converted into NADH models in situ by FDH outside of the capsules. The artificial catalytic and enzymatic processes were isolated in the reaction, allowing them to form relatively independent catalytic processes by virtue of the capsule backbone. In addition, the cofactor channels modified on the capsule eliminate the inherent communication barrier and allow the transport of matter and energy, ensuring that the two reaction systems proceed in parallel and form a closed loop of electrons and protons.

**Catalytic system expansion.** The application scope was explored using NADH-regenerated enzyme mimics to catalyze the monooxygenation of cyclobutanones by FDH. Various cyclobutanone derivatives containing electron-donating or electron-withdrawing groups could be converted to the corresponding products in up to 93% yield (Figs. 6b and 7b). The experiment using a cyclobutanone possessing bulky substituents (3-(4,4″-di-*tert*-butyl-[1,1′:3′,1″-terphenyl]-5′-yl)cyclobutan-1-one, **9a**) that made it larger than the pocket of Zn–**ZPA**, yielded 18% of product **9b** under the same reaction conditions as were used for **1a**, indicating that the monooxygenation reaction occurred inside the pocket of Zn–**ZPA**. A comparison of yields with **1a** in the presence of ATP suggests that the ~18% yield for **9b** was the background of reaction under the reaction condition via the normal homogeneous. At the same time, we confirmed the broad biocompatibility of Zn–**ZPA**, an NADH-regenerating enzyme

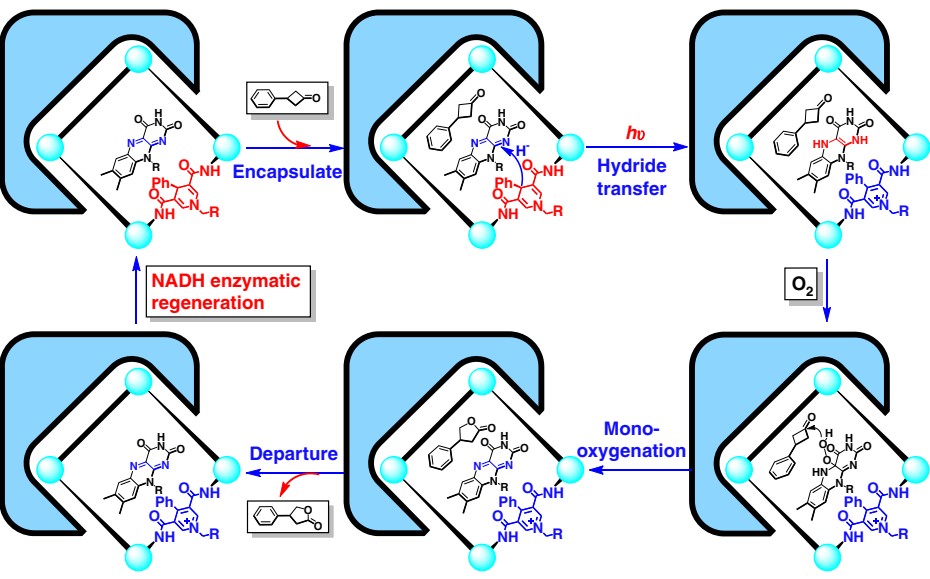

**Fig. 6 Proposed mechanisms for the catalyst–enzyme coupled system in catalytic monooxygenation.** Schematic of the supramolecular approach for the monooxygenation of cyclobutanones showing the enzymatic behavior within the confined environment and in situ regeneration of the NADH model under redox enzymatic catalysis.

simulation platform, by extending it to other enzymes. Comparing the transformation from $HCOO^-$ to $CO_2$ to obtain higher value-added fine chemicals such as $CH_3CHO$ from $CH_3CH_2OH$, the enzymatic systems were driven by alcohol dehydrogenase (ADH)[56]. $CH_3CH_2OH$ ($E_{EtOH/CH3CHO} = 0.84$ V vs Ag/AgCl)[57] displayed a weaker reducing power than $HCOO^-$ ($E_{CO2/HCOO^-} = -0.81$ V vs Ag/AgCl)[58], implying that ADH offered slower catalytic dynamics when combined with artificial catalytic systems relative to the dynamics of FDH. The concentration of ADH in the catalytic system was adjusted to match the reaction processes catalyzed by the host–guest complex to regenerate the NADH models instead of FDH. The catalysis with ADH (120 U mL$^{-1}$) gave **1b** in 78% yield (Fig. 7), which meant that the different catalytic properties of the enzymes with different catalytic dynamics had a substantial impact on the overall catalysis with combinations of enzymes and artificial catalysts. The catalytic performances of both enzymes were also dependent on the formation of host–guest species, and the artificial cofactors in Zn–**ZPA** could facilitate different types of enzymatic reactions, resembling those in living organisms and achieving the desired substrate conversions (Supplementary Fig. 35).

Achieving oxidative desulfurization with high selectivity to produce sulfoxide as an auxiliary reagent is essential in the biomedical and chemical industries[59,60]. Natural flavin-dependent enzymes can metabolize endogenous compounds containing heteroatoms such as sulfur and offer both high selectivity and high efficiency[27]. The superiority of such combined systems that use clean energy and enzymes to promote monooxygenation could be extended to thioethers with high efficiency and selectivity while avoiding overoxidation (Fig. 7). Moreover, the Co(acac)$_3$ as a quencher of singlet oxygen was added the aforementioned catalytic system, which did not result in significant decrease of the yield under the same reaction conditions, further indicating that the use of a weak light can avoid the disturbance of background response from formed singlet oxygen by RFT sensitization[61]. When the loadings of Zn–**ZPA** and RFT were fixed at 2 mol% of the substrates for the reaction with thioether derivatives, Zn–**ZPA** showed higher

activity in the oxidation of thioethers to corresponding sulfoxides with a product yield of ~95% within 4 h, and no sulfones were detected when using different enzymes.

## Discussion

In summary, we have demonstrated a host–guest supramolecular strategy to combine artificial catalyst and enzymes within the domain of enzymes, eliminating the inherent communication barrier and interference between abiotic and biotic systems for effective catalysis based on NADH-modified capsules. The direct introduction of NADH mimics as transport channels for protons and electrons on the surface of a metal-organic host allowed the direct proton and electron transport between the two enzymatic processes inside the pocket of the enzymes without consuming cofactors. And it represents a distinguished example to integrate the cofactor, catalyst and reactant within a confined space for effective catalysis, which is an attempt for combining artificial and natural catalyst. The participation of different NADH-dependent oxidoreductases (for example, FDH and ADH) and substrates (for example, cyclobutanones and thioethers) in the combined artificial catalysis-enzymatic catalysis system with high efficiency and precise selectivity suggested the superiority of this supramolecular approach based on the electron and proton transmission channels of the capsules and the effect of spatial isolation, providing a host–guest catalytic platform for developing scalable and sustainable biocatalysis.

## Methods

**General methods and materials**. Unless stated otherwise, all chemicals and solvents were of reagent grade quality obtained from commercial sources and used without further purification. $^1$H NMR was measured on a Bruker 400 M spectrometer with chemical shifts reported as ppm (in DMSO-$d_6$ or CDCl$_3$, TMS as internal standard). ESI mass spectra were performed on a HPLC-Q-Tof MS spectrometer using methanol as mobile phase. Isothermal Titration Calorimetry (ITC) was performed on a Nano ITC (TA Instruments Inc. Waters LLC). The elemental analyses of C, H and N were performed on a Vario EL III elemental analyzer. UV-Vis spectra were measured on a HP 8453 spectrometer. The solution fluorescent spectra were measured on Edinburgh FS-920. The excitation and emission slits were both 2.5 nm wide. All electrochemical measurements were

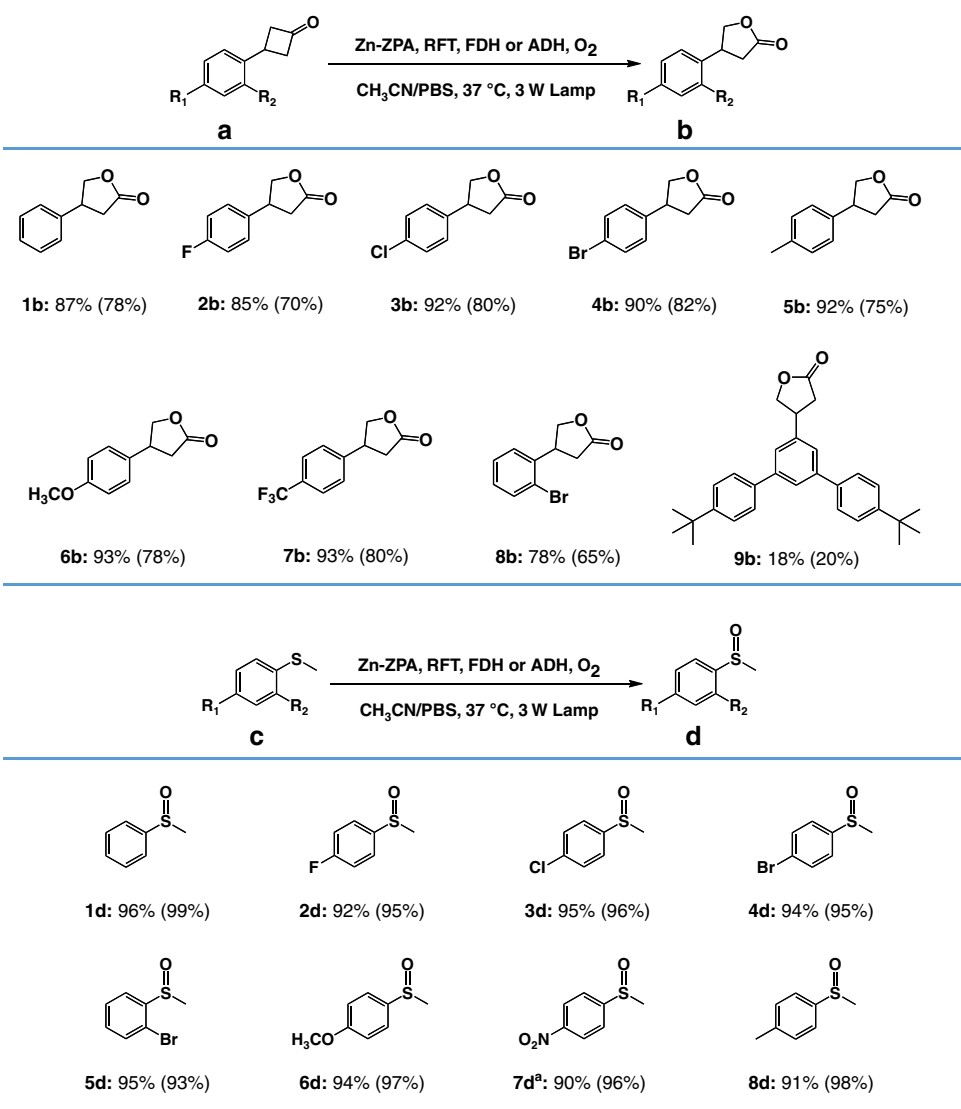

**Fig. 7 Scope of the monooxygenation under light irradiation.** For substituted phenylcyclobutanone: substrate (8.0 mM), Zn–**ZPA** (1.6 mM), RFT (1.6 mM), FDH (2 U mL$^{-1}$) and HCOONa (16.0 mM) in a CH$_3$CN/PBS solution (2:1), 3 W fluorescent lamp, 18 h. The yields obtained with ADH (120 U mL$^{-1}$) and CH$_3$CH$_2$OH (3% in volume) instead of FDH and HCOONa are shown in parentheses. For substituted thioanisole: substrate (8.0 mM), Zn–**ZPA** (0.16 mM), RFT (0.16 mM), FDH (1 U mL$^{-1}$) and HCOONa (16.0 mM) in a CH$_3$CN/PBS solution (2:1), 3 W fluorescent lamp, 4 h. The yields obtained with ADH (60 U mL$^{-1}$) and CH$_3$CH$_2$OH (3% in volume) instead of FDH and HCOONa are shown in parentheses. The yields were determined by $^1$H NMR analysis of the products. [a]12 h.

carried out under argon gas at room temperature performed on a ZAHNER ENNIUM electrochemical workstation with a conventional three-electrode system with a homemade Ag/AgCl electrode as a reference electrode, a platinum silk with 0.5 mM diameter as a counter electrode, and glassy carbon electrode as a working electrode. Products were purified by flash column chromatography on 200–300 mesh silica gel SiO$_2$.

**Preparation of Zn–ZPA.** Zn(BF$_4$)$_2$·H$_2$O (25.7 mg, 0.10 mmol) and H$_2$**ZPA** (58.4 mg, 0.10 mmol) were dissolved in CH$_3$CH$_2$OH/CH$_3$CN (1:1 in volume) to give a yellow solution. The X-ray quality yellow block crystals were grown by diffusing diethyl ether into the solution. Yield: 60%. $^1$H NMR (400 MHz, DMSO-$d_6$, ppm): δ 11.78 (s, 2H), 8.56–8.48 (m, 3H), 8.27 (s, 2H), 8.01 (d, $J$ = 8.4 Hz, 2H), 7.89 (s, 6H), 7.70 (d, $J$ = 8.4 Hz, 2H), 7.42–7.24 (m, 6H), 7.18–7.14 (t, $J$ = 7.2 Hz, 1H), 5.33 (s, 1H), 2.80 (d, $J$ = 4.0 Hz, 3H). Elemental analysis calcd for Zn$_4$C$_{132}$H$_{110}$N$_{32}$O$_{12}$B$_6$F$_{24}$·CH$_3$CN: H, 4.07; C, 57.57; N, 3.51%. Found: H, 4.01; C, 57.73; N, 3.53%. ESI-MS: $m/z$: 864.5278 [H$_3$Zn$_4$(**ZPA**)$_4$]$^{3+}$.

**Preparation of Zn–PMA.** Zn(BF$_4$)$_2$·H$_2$O (25.7 mg, 0.10 mmol) and H$_2$**PMA** (50.5 mg, 0.10 mmol) were dissolved in CH$_3$CH$_2$OH/CH$_3$CN (1:1 in volume) to give a yellow solution. The X-ray quality yellow block crystals were grown by diffusing diethyl ether into the solution. Yield: 52%. $^1$H NMR (400 MHz, DMSO-$d_6$, ppm): δ 12.32 (s, 2H), 8.65–8.48 (m, 8H), 8.05–7.93 (m, 8H), 7.47–7.45 (m, 2H),

2.83 (d, $J$ = 4.0 Hz, 3H). Elemental analysis calcd for Zn$_4$C$_{112}$H$_{92}$N$_{28}$O$_{12}$: H, 4.06; C, 58.91; N, 17.17%. Found: H, 4.14; C, 58.85; N, 17.13%. ESI-MS: $m/z$: 759.14 [H$_3$Zn$_4$(**PMA**)$_4$]$^{3+}$, 1138.17 [H$_2$Zn$_4$(**PMA**)$_4$]$^{2+}$.

**General methods for the catalytic Baeyer–Villiger reactions.** RFT (1.6 mM) combined with HCOONa (16.0 mM) or CH$_3$CH$_2$OH (3% in volume) was added to a CH$_3$CN/PBS solution (2:1, pH 7.40, 5 mL) containing Zn–**ZPA** (1.6 mM) with a magnetic stir bar. A second stock solution of FDH (1 U mL$^{-1}$) or ADH (60 U mL$^{-1}$) in buffer solution was added to the above reaction vial, and then the substrate was added. The oxygen atmosphere in the reaction vial was maintained with an oxygen balloon, and a constant-temperature water circulation device maintained the reaction temperature at 37 °C. The vial was irradiated with a fluorescent lamp (3 W) with constant stirring for 9 h. At this point, another portion of enzyme (the same amount as before) was added, and the reaction was stirred for another 9 h. $^1$H NMR spectra were recorded to determine the yields (relative to the remaining starting material or mesitylene benzene as an internal standard). The crude product could be purified by column chromatography on silica gel to afford the corresponding products.

The general method for the catalytic monooxygenation of thioethers was similar to that of the Baeyer–Villiger reactions; the substrate was changed, the concentrations of Zn–**ZPA** (0.16 mM) and FDH (2 U mL$^{-1}$)/ADH (60 U mL$^{-1}$), were different, and the reaction time was shorter.

**X-ray crystallography**. The intensities were collected on a Bruker SMART APEX CCD diffractometer equipped with a graphite-monochromated Mo-Kα ($\lambda$ = 0.71073 Å) radiation source; the data were acquired using the SMART and SAINT programs[62,63]. The structures were solved by direct methods and refined on $F_2$ by full-matrix least-squares methods using the SHELXTL version 5.1 software[64]. In the structural refinement of Zn–ZPA and Zn–PMA, all the non-hydrogen atoms were refined anisotropically. Hydrogen atoms within the ligand backbones were fixed geometrically at calculated distances and allowed to ride on the parent non-hydrogen atoms. To assist the stability of refinements, one tetrafluoroborate anion and some benzene rings were restrained as idealized regular polygons and the thermal parameters on adjacent atoms in this anion and some benzene rings were restrained to be similar. The SQUEEZE subroutine in PLATON was used[65].

**Crystal data for Zn-ZPA**. $Zn_4C_{132}H_{110}N_{32}O_{12}\cdot6BF_4\cdot4CH_3CH_2OH\cdot8H_2O$, $M$ = 3447.26, Tetragonal, space group $P\bar{4}n2$, yellow block, $a = b = 23.3087(12)$ Å, $c = 16.0801(16)$ Å, $V = 8736.2(11)$ Å$^3$, $Z = 2$, $D_c = 1.310$ g cm$^{-3}$, $\mu$(Mo-Kα) = 0.637 mm$^{-1}$, $T = 180(2)$ K. 7675 unique reflections [$R_{int} = 0.0783$]. Final $R_1$ [with $I > 2\sigma(I)$] = 0.0860, $wR_2$ (all data) = 0.2230 for the data collected. CCDC number 1920570.

**Crystal data for Zn-PMA**. $Zn_4C_{112}H_{84}N_{28}O_{12}\cdot6H_2O$, $M$ = 2383.64, Tetragonal, space group $P\bar{4}$, yellow block, $a = b = 16.4944(7)$, $c = 14.3438(8)$ Å, $V = 3902.4(4)$ Å$^3$, $Z = 1$, $D_c = 1.014$ g cm$^{-3}$, $\mu$(Mo-Kα) = 0.664 mm$^{-1}$, $T = 180(2)$ K. 9997 unique reflections [$R_{int} = 0.1144$]. Final $R_1$ [with $I > 2\sigma(I)$] = 0.0980, $wR_2$ (all data) = 0.2442 for the data collected. CCDC number 1920627.

## Data availability

The X-ray crystallographic coordinates for the structures reported in this article have been deposited at the Cambridge Crystallographic Data Centre (CCDC) under the deposition numbers CCDC 1920570 and 1920627. These data can be obtained free of charge from The Cambridge Crystallographic Data Centre via http://www.ccdc.cam.ac.uk/data_request/cif. All other data supporting the findings of this study are available within the Article and its Supplementary Information files or from the corresponding author upon request.

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

## Acknowledgements

This work was supported by the National Natural Science Foundation of China (Grants Numbers: 21820102001, 21890381 and 21531001).

## Author contributions

L.Z. and J.K.C. contributed equally to this work. L.Z. and C.Y.D. conceived the project, designed the experiments and supervised the work. J.K.C. carried out the main experiments, collected and interpreted the data. Y.N.L. prepared the ligand. L.Z., J.K.C. and J.W.W. solved and refined the X-ray single-crystal structures. L.Z. and C.Y.D. contributed materials and analysis tools. L.Z., J.K.C. and C.Y.D. cowrote the paper. All authors discussed the results and commented on the manuscript.

## Competing interests

The authors declare no competing interests.
