## [Peer Review File · Nature Communications]

Reviewers' comments:

Reviewer #1 (Remarks to the Author):

This work by Zhao et al. reported a cooperative approach of enzymatic and artificial catalysis to catalyze biomimetic monooxygenation. The relevant NADH mimicking work has been reported recently by the same group (J. Am. Chem. Soc. 2019, 141, 12707–12716) and this work they made specific modification to the organic ligand but investigate different reaction models. The supramolecular capsule of metal zinc and ZPA provides a macrocycle pocket and the ligand ZPA works as NADH-mimicking active site which contains a dihydropyridine amido (DHPA) and is similar to the active site of cofactor NADH. Typically, only those NADH mimics distributed on the surface of supramolecular Zn-ZPA will directly interact with enzyme FDH/ADH, which both have cofactor NADH involved in the enzymatic catalysis and will recycle NADH mimics. Also, the pendant CO-NH groups on the ligands could provide potential hydrogen bonding sites and furtherly mimic enzyme behavior during the reaction. This work is well-organized and interesting, and it gives new insights to design new materials to mimic and participate into the biocatalytic process which is still a challenging area. This reviewer would recommend its publishing in Nature Communication if the following issues can be clearly addressed:

1. How is the stability of this metal-organic capsule in PBS buffer? Does the integrity of the Zn-ZPA change before and after the reaction? It only shows the integrity in CH₃CN solution.
2. The authors claimed that the encapsulation of the RFT in the cavity of Zn-ZPA/MPA is theoretically proposed by simply comparing the size of cavity and RFT. This should be experimentally proved.
3. The enzymatic catalysis needs to be conducted under very mild conditions, but this work use co-solvent of CH₃CN/H₂O (v/v 2/1). Usually, most enzymes become almost totally inactive at an organic cosolvent concentration of 60–70% (v/v) (ACS Catal. 2013, 3, 2823–2836). How was the performance of FDH/ADH influenced under proposed conditions?
4. Additional experiments should be conducted to identify the intermediates of reactive oxygen species during the monooxygenation processes.
5. Experimental evidence should be provided to prove the encapsulation of substrate molecules into the cavity during catalysis processes.

Reviewer #2 (Remarks to the Author):

In the manuscript, Duan et al present new interesting approach towards coupled enzymatic-biomimetic catalysis based on capsuling cofactors/enzyme/substrate into metal-organic host. It is demonstrated on flavin-dependent monooxygenation using FAD model and NADH model, which were proved to provide Bayer-Villiger oxidation reaction on a model substrate – substituted cyclobutanone. It is usually difficult to prove function of such complex biomimetic, especially on cyclobutanone oxidation as it is very easy substrate. Nevertheless, participation of all components is clearly proved by authors. Based on the results, a mechanism is proposed. Formation of the reduced flavin is clearly demonstrated by UV-VIS and NMR experiment under argon. Formation of flavin hydroperoxide is also confirmed by UV-VIS. Further, many experiments were done on proving interactions between flavin-NADH moiety. 1/ Unfortunately necessity of light was not proved! It should be mentioned that also ground-state flavin can be reduced by NADH. Such reaction is not fast but in a complex, hydride transfer could be significantly accelerated (König, B. et al: Chiral NADH model systems functionalized with Zn(II)-cyclen as flavin binding site. Tetrahedron 61, 5241-5251, (2005)). 2/ There is substrate scope study added at the end of the manuscript. No doubt about experiments with cyclobutanone derivatives. On the other hand, on my own, oxidation of sulfides to sulfoxides shown in Table 2 is not suitable model reaction to demonstrate the presented coupled system. It has been demonstrated that RFT itself provides aerobic sulfoxidation reaction by both singlet oxygen and electron transfer mechanism (Cibulka, R. et al: Efficient Metal-Free Aerobic Photooxidation of Sulfides to Sulfoxides Mediated by a Vitamin

B2Derivative and Visible Light. *Adv. Synth. Catal.* 358, 1654-1663, (2016)). For the presented system (if it is really light-dependent – see note 1), it would be difficult to recognize the mechanism. On sulfoxidation, singlet oxygen (formed by RFT sensitization) surely participate. For electron rich substrates (not for nitrothioanisole) electron transfer to the excited RFT followed by oxygenation contributes as well.

In summary I believe the topic is interesting for broad community of readers of *Nature Comm.* The manuscript is clearly written and all experiments are well described. Nevertheless from my point of view, there are two important questions/notes (see notes 1 and 2 above) which should be addressed before acceptance of the manuscript.

Minor points:

3/ What about the structure of flavin hydroperoxide. Under light, also other derivatives could be present as demonstrated in Roithova, J. et al: *Flavinium Catalysed Photooxidation: Detection and Characterization of Elusive Peroxyflavinium Intermediates.* *Angew Chem Int Ed Engl* 58, 15412-15420, (2019)).

4/ Interesting paper on flavin encapsulation should be cited in introduction: Chevalier, Y. et al. *Aerobic Baeyer-Villiger Oxidation Catalyzed by a Flavin-Containing Enzyme Mimic in Water.* *Angew Chem Int Ed Engl* 57, 16412-16415, (2018).

5/ typos: Biomimetic (page 1, line 12)

Reviewer #3 (Remarks to the Author):

This manuscript reports on a very interesting catalyst system that is proposed to proceed through the synergistic merging of a supramolecular catalyst and an enzyme. The results are very promising and should be considered for publication in *Nature Communications* after considering the following:

(1) . The reaction is proposed to involve interaction of the RFT with the supramolecular structure through H-bonding. The authors provide reasonable evidence for the H-bonding; but it is not entirely clear that the NADH model is inside the assembly. Given that the authors have the structure of the assembly, it would be good to see a theoretical 'docking study' of the NADH analogs to see if they are indeed located inside the assembly

(2) . Similarly, the authors propose that along with the RFT, the lactone resides in the assembly. A admittedly superficial inspection of the cavity suggests that there is not enough space to entirely encapsulate both of these molecules. Along these lines ketone 19b undergoes oxidation in 18% yield. How is this possible if it can't fit inside? Some explanation for this result, perhaps including a series of background (catalyst deletion) experiments, is required.

(3) There are some questions about the turnover in this system. 1.6mM Zn-ZPA is equivalent to 6.4mM of NADH analog (4 per cluster). The experiments are conducted on 8mM of substrate; therefore even without turnover, one might expect an 80% yield. The authors obtain yields in the range of 78-92%. Is this system really catalytic? An experiment with lower loadings (or increased amounts of substrate) is required to truly get a sense of the catalyst turnover number.

(4) .The authors suggest (or at least Figure 5 does) that the assembly is almost entirely bound by the enzyme? Can the authors truly distinguish this from only part of the assembly being in the enzyme active site. Some discussion of the flexibility and size of the enzyme active site is needed here; especially since major modification to the native enzyme were not clearly required.

(5) The products (lactones and sulfoxides) are both chiral. Given that the enzyme is clearly capable of enantioselective reactions, and the assembly can adopt a chiral conformation by virtue of its octahedral centers, one might expect some enantioselectivity in these reactions. This should be checked. Indeed, enantioselective in the dual catalyst system (along with absence in the cluster

only reaction) would further support the interaction of enzyme with the cluster in the enantiodetermining event.

In closing, this is a very intriguing study that without a doubt has the potential to rise to the level of Nature Communication. The authors are encouraged to address the points raised above to further strengthen this work.

For Reviewer #1

Comments: This work by Zhao et al. reported a cooperative approach of enzymatic and artificial catalysis to catalyze biomimetic monooxygenation. The relevant NADH mimicking work has been reported recently by the same group (J. Am. Chem. Soc. 2019, 141, 12707–12716) and this work they made specific modification to the organic ligand but investigate different reaction models. The supramolecular capsule of metal zinc and ZPA provides a macrocycle pocket and the ligand ZPA works as NADH-mimicking active site, which contains a dihydropyridine amido (DHPA) and is similar to the active site of cofactor NADH. Typically, only those NADH mimics distributed on the surface of supramolecular Zn-ZPA will directly interact with enzyme FDH/ADH, which both have cofactor NADH involved in the enzymatic catalysis and will recycle NADH mimics. Also, the pendant CO-NH groups on the ligands could provide potential hydrogen bonding sites and furtherly mimic enzyme behavior during the reaction. This work is well-organized and interesting, and it gives new insights to design new materials to mimic and participate into the biocatalytic process which is still a challenging area. This reviewer would recommend its publishing in Nature Communication if the following issues can be clearly addressed:

1. How is the stability of this metal-organic capsule in PBS buffer? Does the integrity of the Zn-ZPA change before and after the reaction? It only shows the integrity in CH₃CN solution.

Response: Thanks very much for the careful suggestions. As the mentioned by the reviewer, the stability of metal-organic capsule in CH₃CN solution before the catalytic reaction has been showed in the manuscript. Due to the poor solubility of Zn-ZPA in absolute PBS buffer, it is difficult to detect the stability of Zn-ZPA in PBS buffer. Our ionization-mass spectrometry of the Zn-ZPA capsule in CH₃CN/PBS mixed solvent exhibited that the peak at approximately $m/z = 864.52$ attributed to Zn-ZPA could be detected both before and after the reaction. It is postulated that the integrity of the Zn-ZPA did not change before and after the reaction. We added some words about the aforementioned result in the revised manuscript.

2. The authors claimed that the encapsulation of the RFT in the cavity of Zn-ZPA/MPA is theoretically proposed by simply comparing the size of cavity and RFT. This should be experimentally proved.

Response: Many thanks to the reviewer for the suggestion. The host-guest behavior between the Zn-ZPA/PMA and RFT is the cornerstone of catalysis and has been well characterized in the manuscript. First: we have used electrospray ionization-mass spectrometry to characterize the encapsulation of the RFT in the supramolecular hosts. As shown in Fig. 4c and Supplementary

Fig. 4.2, the addition of RFT (0.1 mM) into the solution of Zn-ZPA/PMA (0.1 mM) caused the appearance of two intense peaks at $m/z = 1046.5880$ and 1411.2938 attributed to $[\text{H}_3\text{Zn}_4(\text{ZPA})_4(\text{RFT})]^{3+}$ and $[\text{H}_2\text{Zn}_4(\text{PMA})_4(\text{RFT})]^{2+}$, respectively, demonstrating the formation of a 1:1 stoichiometric host-guest complex species $\text{Zn-ZPA/PMA} \supset \text{RFT}$ in solution. Second: the isothermal titration calorimetry showed a 1:1 ratio for Zn-ZPA/RFT host-guest species with a Gibbs free energy change of $\Delta G = -27.39 \text{ kJ}\cdot\text{mol}^{-1}$ and an association constant of $K_a = 6.23 \times 10^4 \text{ M}^{-1}$ for the Zn-ZPA \supset RFT complex (Fig. 4b and Supplementary Fig. 5.16). Third: the addition of Zn-ZPA (10.0 mM) to RFT (10.0 mM) caused the ^1H NMR signals associated with active protons H_z on RFT and H_h on Zn-ZPA shifted downfield, further confirming that the formation of multiple hydrogen bonds between RFT and Zn-ZPA (Fig. 3b)⁴³ shortened the distance between active sites effectively. These experimental results have been mentioned in the manuscript.

3. The enzymatic catalysis needs to be conducted under very mild conditions, but this work use co-solvent of $\text{CH}_3\text{CN}/\text{H}_2\text{O}$ (v/v 2/1). Usually, most enzymes become almost totally inactive at an organic cosolvent concentration of 60–70% (v/v) (ACS Catal. 2013, 3, 2823–2836). How was the performance of FDH/ADH influenced under proposed conditions?

Response: Thanks very much for the thoughtful suggestion. In fact, we have realized that mixed solvent systems are not the best conditions for enzymatic reactions, but the catalytic solubility of metal-organic host, enzymes and reactants should be considered and the solvent system was finally determined as $\text{CH}_3\text{CN}/\text{H}_2\text{O}$ (v/v 2/1). In order to show the influence of mixed solvent systems, we have conducted controlled experiments with different water content systems. The results showed that the increase of water content to $\text{CH}_3\text{CN}/\text{H}_2\text{O}$ (v/v 1/1) and decrease of water content to $\text{CH}_3\text{CN}/\text{H}_2\text{O}$ (v/v 3/1) gave a yield of 52% and 21% for **1b**, respectively, suggesting that the water content affect the activity of enzymes and the solubility of substrates and capsules, which inhibited the occurrence of reaction. We have noted that the supramolecular approach played a significant role in the enzymatic catalysis performed in mixed solvent system, the conduction of the enzymatic reaction in aqueous system by water-soluble capsules would have a better consequence.

4. Additional experiments should be conducted to identify the intermediates of reactive oxygen species during the monooxygenation processes.

Response: Thanks very much for the thoughtful suggestion. We have identified the formation of the intermediates of reactive oxygen species by UV-Vis spectrum during the monooxygenation processes. In the light of the reviewer' suggestion, we added some additional experiments to

identify this point. (1) As mentioned in manuscript, ^1H NMR monitoring of the hydride transfer processes between the RFT and Zn-ZPA in the absence of oxygen showed a gradual increase of RFTH_2 (Fig. 3d). When exposing aforementioned solution to the oxygen, the peaks attributable to RFTH_2 disappeared and a new peak formed at approximately 10.2 ppm, implying the formation of peroxide species in the system (Supplementary Fig. 5.4). (2) The potassium iodide starch test paper was used to visually detect the peroxide in the reaction system. It could be seen that the potassium starch iodide test paper turned blue quickly when it came in contact with the reaction solution (Supplementary Fig. 5.21)⁴⁵. (3) The reaction intermediate attributed to Zn-ZPA \supset RFTOOH at $m/z = 1046.2420$ was able to be detected by electrospray ionization mass spectrometry, which further verified the generation of peroxide during the reaction. To make this point more clearly, we have added some word to identify the intermediates of reactive oxygen species in the revised the manuscript. And supplementary information for ^1H NMR monitoring and potassium iodide starch test were added in the revised version supporting information and referred in corresponding place of the revised manuscript.

5. Experimental evidence should be provided to prove the encapsulation of substrate molecules into the cavity during catalysis processes.

Response: Thanks very much for the thoughtful suggestion. The direct experimental evidence to prove the encapsulation of substrate molecules into the cavity during catalysis was come from the electrospray ionization-mass spectrometry results. As shown in Fig. 4c, the simultaneous addition of the substrate (3-phenylcyclobutanone, **1a**, 0.1 mM) and RFT (0.1 mM) into the solution of Zn-ZPA (0.1 mM) caused the appearance of a new peak at $m/z = 1095.2858$, which was assigned to 1:1:1 host-catalyst-substrate species $[\text{H}_3\text{Zn}_4(\text{ZPA})_4(\text{RFT})(\mathbf{1a})]^{3+}$. On the other hand, the pseudo-zero-order kinetic behaviour, size selectivity and reaction inhibition that can be accepted for the encapsulation of substrate molecules into the cavity during catalysis processes. In order to prove this point, we carried out some control experiments in this article. (1) The catalytic tracing experiment showed a pseudo-zero-order kinetic behavior, which was consistent with a Michaelis-Menten mechanism (Fig. 5a). (2) We tried a large substrate with bulky substituents (Table 2, **9b**), and the experimental results showed that a much lower yield of corresponding products generated. (3) An inhibition experiment with ATP as an inhibitor was performed to give a low yield under the same reaction conditions (Supplementary Table 6.1, entry 3). These experimental evidences were supposed to prove the encapsulation of substrate molecules into the cavity during catalysis processes.

For Reviewer #2

Comments: In the manuscript, Duan et al present new interesting approach towards coupled enzymatic-biomimetic catalysis based on capsuling cofactors/enzyme/substrate into metal-organic host. It is demonstrated on flavin-dependent monooxygenation using FAD model and NADH model, which were proved to provide Bayer-Villiger oxidation reaction on a model substrate – substituted cyclobutanone. It is usually difficult to prove function of such complex biomimetic, especially on cyclobutanone oxidation as it is very easy substrate. Nevertheless, participation of all components is clearly proved by authors. Based on the results, a mechanism is proposed. Formation of the reduced flavin is clearly demonstrated by UV-VIS and NMR experiment under argon. Formation of flavin hydroperoxide is also confirmed by UV-VIS. Further, many experiments were done on proving interactions between flavin-NADH moiety.

In summary, I believe the topic is interesting for broad community of readers of Nature Comm. The manuscript is clearly written and all experiments are well described. Nevertheless, from my point of view, there are two important questions/notes (see notes 1 and 2 above) which should be addressed before acceptance of the manuscript.

1. Unfortunately necessity of light was not proved! It should be mentioned that also ground-state flavin can be reduced by NADH. Such reaction is not fast but in a complex, hydride transfer could be significantly accelerated (König, B. et al: Chiral NADH model systems functionalized with Zn(II)-cyclen as flavin binding site. Tetrahedron 61, 5241-5251, (2005)).

Response: Thanks very much for the thoughtful suggestion. One important point should be noted was that the hydride transfer process from NADH mimics to RFT is the first step, and is far from the rate-determining step of entire monooxygenation cyclic reaction in our system. The reaction rate of the hydride transfer and other steps before the monooxygenation step of the substrate by the active oxygen species should be higher enough to ensure subsequent reactions. As the mentioned by the reviewer, we have noted that ground-state flavin can indeed be reduced by NADH and the hydride transfer can be improved through supramolecular approaches. Nevertheless, the reaction rate of the whole monooxygenation of our supramolecular system was comparable to that of the fastest hydride transfer reported by König. It is postulated that the hydride transfer from NADH mimics to the ground-state flavin was not fast enough to efficiently promote the subsequent monooxygenation reaction. In fact, in our control experiments, only trace target products were detected even if the reaction time was extended to 60 h in the absence of light, further implying that the use of light was required for further accelerating the hydride

transfer in our reaction system. Actually, we have tested ^1H NMR and UV-Vis spectrum of Zn-ZPA and RFT under the irradiation at 455 nm, showing the rapid and improved hydride transfer processes. So that the introduction of light was an ideal way to promote the reactions, and we could achieve reactions with just a 3 W household fluorescent lamp.

2. There is substrate scope study added at the end of the manuscript. No doubt about experiments with cyclobutanone derivatives. On the other hand, on my own, oxidation of sulfides to sulfoxides shown in Table 2 is not suitable model reaction to demonstrate the presented coupled system. It has been demonstrated that RFT itself provides aerobic sulfoxidation reaction by both singlet oxygen and electron transfer mechanism (Cibulka, R. et al: Efficient Metal-Free Aerobic Photooxidation of Sulfides to Sulfoxides Mediated by a Vitamin B₂ Derivative and Visible Light. *Adv. Synth. Catal.* 358, 1654–1663, (2016)). For the presented system (if it is really light-dependent – see note 1), it would be difficult to recognize the mechanism. On sulfoxidation, singlet oxygen (formed by RFT sensitization) surely participate. For electron rich substrates (not for nitrothioanisole) electron transfer to the excited RFT followed by oxygenation contributes as well.

Response: Many thanks to the reviewer for the thoughtful suggestion. Actually, we have realized that RFT itself provided aerobic sulfoxidation reaction by singlet oxygen, in the meanwhile, we also noted that the occurrence of aerobic sulfoxidation reaction required the irradiation with higher power light (for example 450 nm 7 W diodes in this reference). To avoid the disturbance of singlet oxygen in the catalysis, a weaker light was selected (3 W fluorescent lamp) in our researches. For better clarifying the role of light and our supramolecular system plays in it, we performed the following control experiments. Typically, irradiation (7 W LED light with 455 nm) of a CH₃CN/PBS (2:1) solution containing electron rich (4-bromophenyl) (methyl)sulfane (8.0 mM), RFT (0.16 mM), formate dehydrogenase (FDH, 1 U·mL⁻¹) and HCOONa (16.0 mM) afforded product sulfoxides in a high yield with or without Zn-ZPA (0.16 mM), suggesting that this system could efficiently complete the catalytic conversion of sulfide under illumination with high power light. However, when substituting 3 W fluorescent lamp for 7 W LED light with 455 nm, two yields (94% and 6%) that have a very sharp contrast between catalytic systems in the presence and absence of Zn-ZPA were observed, indicating that our supramolecular system indeed offers the key advantages in integrated chemical and biological synthetic sequences, although there still have a small amount of product given by singlet oxygen or/and electron transfer mechanism. Moreover, according to this reference, the Co(acac)₃ as a quencher of singlet oxygen was added the aforementioned catalytic system, which still gave a

yield of up to 85% under the same reaction conditions, further indicating that only a small portion of production *via* the singlet oxygen or electron transfer mechanism under irradiation with a weak light. To make this point more clearly, we have added some word to supplement this statement in the revised the manuscript.

Minor points:

3. What about the structure of flavin hydroperoxide. Under light, also other derivatives could be present as demonstrated in Roithova, J. et al: Flavinium Catalysed Photooxidation: Detection and Characterization of Elusive Peroxyflavinium Intermediates. *Angew Chem Int Ed Engl* 58, 15412–15420, (2019)).

Response: Thanks very much for the careful suggestion. It is generally accepted that the biomimetic monooxygenation proceeds *via* flavin hydroperoxide RFTOOH intermediate^{29,30}. In the light of the reviewer' suggestion, we further performed ¹H NMR and electrospray ionization mass spectrometry (ESI-MS) to characterize the structure of flavin hydroperoxide. Only one new signal peak attributed to peroxide RFTOOH and the reaction intermediate attributed to Zn–ZPA ⇌ RFTOOH at $m/z = 1046.2420$ were captured and other flavin hydroperoxide derivatives were not detected, suggesting that RFTOOH was the main form of peroxyflavinium intermediate. To make this point more clearly, we have added some word to supplement this statement in the revised the manuscript.

4. Interesting paper on flavin encapsulation should be cited in introduction: Chevalier, Y. et al. Aerobic Baeyer-Villiger Oxidation Catalyzed by a Flavin-Containing Enzyme Mimic in Water. *Angew Chem Int Ed Engl* 57, 16412–16415, (2018).

Response: Thanks very much for the careful suggestion. We cited this literature in corresponding site to support our statement in the revised the manuscript and all changes are marked in light yellow.

5. typos: Biomemetic (page 1, line 12)

Response: We are sorry for this careless mistake and many thanks for the careful suggestion. This typo has been corrected in the revised manuscript and full manuscript was carefully checked to correct some typos for improving quality.

For Reviewer #3

Comments: This manuscript reports on a very interested catalysts system that is proposed to proceed through the synergistic merging of a supramolecular catalyst and an enzyme. The results are very promising and should be considered for publication in Nature Communications after considering the following:

1. The reaction is proposed to involve interaction of the RFT with the supramolecular structure through H-bonding. The authors provide reasonable evidence for the H-bonding; but it is not entirely clear that the NADH model is inside the assembly. Given that the authors have a structure of the assembly, it would be good to see a theoretical 'docking study' of the NADH analogs to see if they are indeed located inside the assembly.

Response: Thanks very much for the careful suggestion. In the light of the reviewer's suggestion, we performed the corresponding theoretical 'docking study' for supporting that RFT indeed located inside the assembly. The result showed that supramolecular host modified by the NADH module is large enough to encapsulate the flavin analogue RFT with multiple hydrogen bonds formed between the hydrazides on the capsule Zn-ZPA and the carbonyl group and acetyls of RFT (Supplementary Fig. 7.1). The hydrogen-bond sites widely distributed on the backbone of host and the agglomeration effect between host and guest promote the formation of 1:1 host-guest species, shortening distance between the catalytic centers and cofactors and optimizing the reactions. To make this point more clearly, we have added some word to supplement this statement in the revised the manuscript. And theoretical calculation was added in the revised supporting information and referred in corresponding place of the revised manuscript.

2. Similarly, the authors propose that along with the RFT, the lactone resides in the assembly. An admittedly superficial inspection of the cavity suggests that there is not enough space to entirely encapsulate both of these molecules. Along these lines ketone 9b undergoes oxidation in 18% yield. How is this possible if it can't fit inside? Some explanation for this result, perhaps including a series of background (catalyst deletion) experiments, is required.

Response: Thanks very much for the thoughtful suggestion. As the mentioned by the reviewer, the ketone **9b** with the molecular size larger than the opening of the capsule, undergoes oxidation in 18% yield, however, our control experiment that addition of ATP as a retarder would not quench the reaction significantly. As the competitive inhibition behavior by inhibitor ATP on the catalytic system with substrate **1a** demonstrated a significant quenching of the reaction from the

conversion of 87% of **1b** to that of 18% in the presence of ATP, which suggested that the approximately 18% yield for **9b** was the background of reaction *via* the normal homogeneous. The direct experimental evidence to prove the encapsulation of substrate molecules into the cavity during catalysis processes was come from the electrospray ionization-mass spectrometry result. The simultaneous addition of the substrate (3-phenylcyclobutanone, **1a**, 0.1 mM) and RFT (0.1 mM) into the solution of Zn-ZPA (0.1 mM) caused the appearance (Fig. 4c) of a new peak at $m/z = 1095.2858$, which was assigned to 1:1:1 host-catalyst-substrate species $[H_3Zn_4(ZPA)_4(RFT)(1a)]^{3+}$. The fact that the catalytic experiment showed a pseudo-zero-order kinetic behavior consistent with a Michaelis-Menten mechanism (Fig. 5a), which also supported that the reaction occurred in the cavity of the capsule. To make this point more clearly, we have added some word to explain this result in the revised the manuscript.

3. There are some questions about the turnover in this system. 1.6 mM Zn-ZPA is equivalent to 6.4 mM of NADH analog (4 per cluster). The experiments are conducted on 8mM of substrate; therefore, even without turnover, one might expect an 80% yield. The authors obtain yields in the range of 78-92%. Is this system really catalytic? An experiment with lower loadings (or increased amounts of substrate) is required to truly get a sense of the catalyst turnover number.

Response: Thanks very much for the careful suggestion. In the light of the reviewer' suggestion, we performed a control experiments with the concentration of substrate **1a** was increased from 8.0 mM to 16.0 mM under the typical conditions, the yield of product **1b** increased from 27 μ mol to 45 μ mol (Fig. 5d). When the concentration of substrate increased to 32.0 mM, the yield of product **1b** increased to 78 μ mol. The higher conversion with the lower loading of the catalyst by increasing the substrate amounts clearly revealed a catalytic system. We also performed a control experiment with substrate **1a** (8.0 mM), Zn-ZPA (1.6 mM) and RFT (1.6 mM) in a CH₃CN/PBS solution (2/1, pH 7.40, 5 mL) without enzyme, while only a yield of 15% was observed (Supplementary Table. 6.1). Clearly, these controlled experiments supported that the reaction system is a true catalytic system, some words were added in the revised manuscript to clarify this point in the revised manuscript.

4. The authors suggest (or at least Figure 5 does) that the assembly is almost entirely bound by the enzyme? Can the authors truly distinguish this from only part of the assembly being in the enzyme active site. Some discussion of the flexibility and size of the enzyme active site is needed here; especially since major modification to the native enzyme were not clearly required.

Response: Thanks very much for the careful suggestion. As the mentioned by the reviewer, the

enzymatic catalytic systems are always complex, the conformational changes of the coenzyme on Zn-ZPA during the reaction and the adaptability of the enzyme to the guest molecules consequently make it very hard to truly distinguish the relative positions and interactions between our capsule and the active site in the enzymes. Inspired by the same reviewer, we using theoretical calculations to further clarify the existence of capsules in enzymes by virtue of 'docking study' method. The theoretical 'docking study' suggested that the capsule could enter the enzyme cavity in line with our expectations, despite in a partially hidden way. To make this point more clearly, we have added some words and revised schematic to clarify this point in the revised manuscript, within which the graphics of both capsule and enzyme were abstracted and simplified for aesthetics and clarity in the Figure 5 (which is Figure 6 in revised manuscript).

5. The products (lactones and sulfoxides) are both chiral. Given that the enzyme is clearly capable of enantioselective reactions, and the assembly can adopt a chiral conformation by virtue of its octahedral centers, one might expect some enantioselectivity in these reactions. This should be checked. Indeed, enantioselective in the dual catalyst system (along with absence in the cluster only reaction) would further support the interaction of enzyme with the cluster in the enantiodetermining event.

Response: Thanks very much for the thoughtful suggestion. The capsule Zn-ZPA crystallizes in the tetragonal space group $P\bar{4}n2$, showing a crystallographic C_2 -axis passing through the center of tetranuclear macrocycle. Although its octahedral metal centers adopt a chiral conformation, the inner space of molecular capsule is non-chiral environment. To make this point more clearly, we further tested the products (lactones and sulfoxides) whether have a single chirality. The result indicated that the products are racemic, and all of the conversions were calculated based on the total amount of racemates. The non-chiral products further confirmed that the oxidation reaction actually occurs inside the capsule. We also added some words to clarify this point in the revised manuscript. We think the reviewer's suggestions are very meaningful, and we have started trying to catalyse the chiral transmission by constructing a chiral capsule.

In closing, this is a very intriguing study that without a doubt has the potential to rise to the level of Nature Communication. The authors are encouraged to address the points raised above to further strengthen this work.

REVIEWERS' COMMENTS:

Reviewer #1 (Remarks to the Author):

The authors have satisfactorily addressed all the comments from the reviewer and the manuscript can now be accepted as is.

Reviewer #2 (Remarks to the Author):

Duan et al report flavin-dependent biomimetic coupled system consisting of a metal-organic capsule and an enzyme, formate dehydrogenase or alcoholdehydrogenase. The system has been characterized by several techniques and its function has been proved on B.V oxidation and sulfoxidation reactions. It was confirmed that the processes occur in catalytic regime. The manuscript is clearly written and accompanied by several data in SI. Considering novelty, topic as well as overall quality, I have impression the manuscript is suitable for Nature Communication journal.

Authors did a lot of work and addressed most of suggestions, notes and recommendations of reviewers in the revised manuscript. Nevertheless, there are some minor issues that could be addressed before acceptance. They are mainly important to refine some sentences and conclusions. Authors confirmed stability of Zn-ZPA in acetonitrile – PBS buffer mixtures (question 1, reviewer 1) using Q-TOF-MS analysis analogously as they had done for pure acetonitrile solutions in the original manuscript. They also mentioned this fact in the revised manuscript (the second paragraph of Results). I recommend addition of information that the original data were measured in acetonitrile as it is not stated in the revised text. Thus, note about acetonitrile – PBS mixture in the last sentence looks like without context. Composition of solvent mixtures used for stability experiments should be specified in the text (it shows range of applicability for potential users). In addition to UV-VIS experiments, confirmation of the formation of RFTOOH from RFTH2 and oxygen (question 3, reviewer 1 and question 3, reviewer 2) was done using starch test usually used for this reason, by MS and using 1H NMR. In 1H NMR, new peak 10.2 ppm appeared which is interpreted that it “confirms the formation of RFTOOH species”. While the result of starch test gives clear evidence about formation of peroxide species, NMR experiment could be interpreted with some degree of uncertainty: peak 10.2 in 1H NMR can be interpreted as a signal of flavin hydroperoxide formed. One should keep in mind that there is not standard in the literature and there is no other NMR experiment confirming that peak 10.2 belongs to RFTOOH particle. Comparison of Baeyer-Villiger oxidation in the presence and in the absence of light is the best test to confirm that the reaction is light-driven. Reaction “in dark” has been done by authors as mentioned in the response to question 1, reviewer 2. Importantly, this experiment should be added into Table 1 as the Entry 9 to complete blank experiments.

Reviewer #3 (Remarks to the Author):

The authors have made a good effort to address the reviewer comments. While some questions remain and to the degree of catalysis (especially in the sulfoxidation) occurs within the supramolecular assembly versus the traditional homogeneous catalysis background, it is convincing enough that some catalysis is occurring through encapsulation to merit publication in Nature Comm.

That being said, it would be useful if the authors presented a more accurate depiction of the capsule and the enzyme. The structure shown in Figures S7.2 suggests that only the benzamides of the ZPA ligand interact with the enzyme, while Figure 6 seems to suggest that a large percentage of the capsule resides in a pocket created by the enzyme. This should be changed to more accurately reflect the conclusion of the docking experiment.

For Reviewer #1

Comments: The authors have satisfactorily addressed all the comments from the reviewer and the manuscript can now be accepted as is.

Response: Many thanks to the referee for the positive comments. We will try our best to improve the quality of our publish.

For Reviewer #2

Comments: Duan et al report flavin-dependent biomimetic coupled system consisting of a metal-organic capsule and an enzyme, formate dehydrogenase or alcohol dehydrogenase. The system has been characterized by several techniques and its function has been proved on B.V oxidation and sulfoxidation reactions. It was confirmed that the processes occur in catalytic regime. The manuscript is clearly written and accompanied by several data in SI. Considering novelty, topic as well as overall quality, I have impression the manuscript is suitable for Nature Communication journal.

Authors did a lot of work and addressed most of suggestions, notes and recommendations of reviewers in the revised manuscript. Nevertheless, there are some minor issues that could be addressed before acceptance. They are mainly important to refine some sentences and conclusions.

1. Authors confirmed stability of Zn-ZPA in acetonitrile–PBS buffer mixtures (question 1, reviewer 1) using Q-TOF-MS analysis analogously as they had done for pure acetonitrile solutions in the original manuscript. They also mentioned this fact in the revised manuscript (the second paragraph of Results). I recommend addition of information that the original data were measured in acetonitrile as it is not stated in the revised text. Thus, note about acetonitrile – PBS mixture in the last sentence looks like without context. Composition of solvent mixtures used for stability experiments should be specified in the text (it shows range of applicability for potential users).

Response: We are sorry for this careless omission and many thanks to the reviewer for the kind suggestions. The MS information of Zn–ZPA performed in acetonitrile and the composition of solvent mixtures used for stability experiments have been added in the revised manuscript and refined the related sentences to make this point more clearly.

2. In addition to UV-VIS experiments, confirmation of the formation of RFTOOH from RFTH₂ and oxygen (question 3, reviewer 1 and question 3, reviewer 2) was done using starch test usually used for this reason, by MS and using ¹H NMR. In ¹H NMR, new peak 10.2 ppm appeared which is interpreted that it “confirms the formation of RFTOOH species”. While the result of starch test gives clear evidence about formation of peroxide species, NMR experiment could be interpreted with some degree of uncertainty: peak 10.2 in ¹H NMR can be interpreted as a signal of flavin hydroperoxide formed. One should keep in mind that there is not standard in the literature and there is no other NMR experiment confirming that peak 10.2 belongs to RFTOOH particle.

Response: Thanks very much for the careful suggestion. Actually, we have done our best to measure the formation of RFTOOH species by starch test, MS, UV-Vis and ¹H NMR. However,

as the referee said, the description for peak 10.2 in ^1H NMR should be in some degree of uncertainty. At the same time, we also consulted a lot of literatures, but did not obtain ^1H NMR data relevant to RFTOOH particle. We fully recognize reviewer's generous suggestions for modification. To make this point more clearly, we have added some word to refine this expression in the revised the manuscript.

3. Comparison of Baeyer-Villiger oxidation in the presence and in the absence of light is the best test to confirm that the reaction is light-driven. Reaction “in dark” has been done by authors as mentioned in the response to question 1, reviewer 2. Importantly, this experiment should be added into Table 1 as the Entry 9 to complete blank experiments.

Response: Thanks very much for the careful suggestion. According to the referee's suggestion, the reaction performed in dark as a control have been added into Table 1 as the Entry 8 to complete blank experiments. And we have added supplementary descriptions and chart modifications in the revised manuscript.

For Reviewer #3

Comments: The authors have made a good effort to address the reviewer comments. While some questions remain and to the degree of catalysis (especially in the sulfoxidation) occurs within the supramolecular assembly versus the traditional homogeneous catalysis background, it is convincing enough that some catalysis is occurring through encapsulation to merit publication in Nature Comm.

1. That being said, it would be useful if the authors presented a more accurate depiction of the capsule and the enzyme. The structure shown in Figures S7.2 suggests that only the benzamides of the ZPA ligand interact with the enzyme, while Figure 6 seems to suggest that a large percentage of the capsule resides in a pocket created by the enzyme. This should be changed to more accurately reflect the conclusion of the docking experiment.

Response: Thanks very much for the careful suggestion. As the author pointed out, it would be better to present a more accurate depiction of the capsule and the enzyme and give a more appropriate scheme to reflect the conclusion of the docking experiment. To make the relationship between capsule and the enzyme more clearly, we have added some words and revised schematic to clarify this point in the revised manuscript.